# Practical exact algorithm for trembling-hand equilibrium refinements in games

**Gabriele Farina**
Computer Science Department
Carnegie Mellon University
gfarina@cs.cmu.edu

**Nicola Gatti**
DEIB
Politecnico di Milano
nicola.gatti@polimi.it

**Tuomas Sandholm**
Computer Science Department
Carnegie Mellon University
sandholm@cs.cmu.edu

## Abstract

Nash equilibrium strategies have the known weakness that they do not prescribe rational play in situations that are reached with zero probability according to the strategies themselves, for example, if players have made mistakes. Trembling-hand refinements—such as extensive-form perfect equilibria and quasi-perfect equilibria—remedy this problem in sound ways. Despite their appeal, they have not received attention in practice since no known algorithm for computing them scales beyond toy instances. In this paper, we design an exact polynomial-time algorithm for finding trembling-hand equilibria in zero-sum extensive-form games. It is several orders of magnitude faster than the best prior ones, numerically stable, and quickly solves game instances with tens of thousands of nodes in the game tree. It enables, for the first time, the use of trembling-hand refinements in practice.

## 1   Introduction

*Nash equilibrium (NE)* [Nash, 1950] is the most seminal solution concept in game theory. However, in many games it is too permissive, prescribing unsatisfactory strategies. In the case of imperfect-information extensive-form games, one limitation is that some NEs do not prescribe optimal play after the player or the opponent has made a mistake. Other issues are that some NEs may prescribe non-credible threats or weakly dominated strategies.

Since the classic paper by Selten [1975], *trembles* have played a crucial role in refining—that is, further curtailing—the set of NEs, to address these issues. Intuitively, trembles represent potential mistakes by the players. Refined solutions then are limit points of NEs as the mistake probabilities approach zero (different refinement concepts have different additional constraints on the trembles, which we will make specific in the next sections). The primary role of trembles is to guarantee that the solutions are *sequentially rational* [Kreps and Wilson, 1982]. The two most famous trembling-hand solution concepts that refine NEs while guaranteeing sequential rationality are *quasi-perfect equilibria (QPEs)* [van Damme, 1984] and *extensive-form perfect equilibria (EFPEs)* [Selten, 1975].[1] In a QPE, a player plays optimally at every information set taking into account possible future mistakes of her opponent but assuming that she will not make future mistakes. In an

EFPE, a player also takes her own possible future mistakes into account. Interestingly, EFPEs need not be a subset of QPEs: the two sets can be disjoint [Hillas and Kohlberg, 2002]. (An EFPE may prescribe the players to play weakly dominated strategies, while a QPE never does.)

Sequential rationality guarantees that the strategies in QPEs and EFPEs are optimal also in parts of the game that are reached with zero probability in equilibrium. This is important, for example, when playing against a human player, who is expected not to always play optimally. No practically viable algorithm for computing trembling-hand refinements in large imperfect-information games is known. For instance, consider the recent milestone where an AI, *Libratus* [Brown and Sandholm, 2017b,a,c]—that was created automatically using equilibrium-finding approaches—beat top human specialist professionals in heads-up no-limit Texas hold'em poker. The techniques were based on approximating NE: no NE refinement was used to capitalize on the opponent's mistakes.

Sequential rationality should not be confused with the concept of *undominated* strategy van Damme [1987]. Miltersen and Sørensen [2006, page 108] illustrate the weakness of undominated Nash equilibria (UNEs) and their inability to capitalize on mistakes compared to QPEs. In particular, UNEs do not preclude a player from *hoping* for a "gift" (mistake) from the opponent. Poker illustrates this problem well: only very few strategies are dominated since there is often room for the opponent to make mistakes later. In other words, while restricting to undominated strategies is a step in the right direction, it does not rule out sequentially-irrational behavior for either player. The solution concepts we study in this paper—QPEs and EFPEs—guarantee sequential rationality and are a standard solution to this problem.[2]

## 1.1 Prior state of the art

In two-player zero-sum games, the problems of finding a QPE or an EFPE can be formulated as limit points of linear programs parameterized by the tremble magnitude. In each linear program in that sequence, the trembles are captured by requiring each action to be played with at least some lower-bound probability. Assuming the game is of perfect recall, the linear program can be written in the sequence form [Koller *et al.*, 1996; Romanovskii, 1962; von Stengel, 1996], which we will make precise later in the paper. The trembles used for the two solution concepts are different, leading to problems of different nature. In the case of QPE, the trembles appear in the objective function and in the right-hand side of the constraints [Miltersen and Sørensen, 2010]. In the case of EFPE, the trembles only appear in the constraint matrix [Farina and Gatti, 2017]. Like Nash equilibrium, a QPE and EFPE can be computed in polynomial time in the size of the input game. However, the big-$O$ complexity hides dramatically larger constants in the case of a QPE or an EFPE, and the algorithms known so far thus do not scale beyond small instances [Čermák *et al.*, 2014; Ganzfried and Sandholm, 2015].

Solutions under both concepts can be found by setting the perturbation magnitude $\epsilon$ to a sufficiently small value that guarantees that the basis of the optimal solution with that value remains optimal as $\epsilon \downarrow 0$,[3] and then by solving the corresponding LP by any LP oracle for that sufficiently small $\epsilon$. Miltersen and Sørensen [2010] and Farina and Gatti [2017] provide a formula for such an $\epsilon$ for QPEs and EFPEs, respectively, and show that it is always representable using a polynomial number of bits. However, in practice, calling an LP oracle with that value of $\epsilon$ is impractical because the value is extremely small. This causes fatal numerical instability in the LP solver if finite-precision (i.e., real-valued) arithmetic is used. If rational (i.e., infinite-precision) arithmetic is used, the LP oracle is prohibitively slow.

An alternative algorithm to compute a QPE is a simplex algorithm variant that deals symbolically with the perturbation using the lexico-minimum ratio test [Miltersen and Sørensen, 2010]. That algorithm may require exponential time as it relies on the simplex algorithm, and it was not known if, in practice, it can scale up to large instances. Our experiments show that it does not. While in principle also an EFPE can be computed using a simplex algorithm that deals with the perturbation symbolically, it is not even clear whether it can run in polynomial space. In summary, although there

---

action must be. Therefore, the trembles are a function of the strategies of the players. This potentially complicates equilibrium finding. It is unknown whether PEs can be efficiently found in extensive-form games.

[2]Čermák *et al.* [2014] show experimentally that in two small artificial poker variants (six cards in the deck and two betting rounds), for two particular models of opponent mistakes, undominated equilibria are as good as QPEs. As explained above, this is not the case in general.

[3]We use the symbol "↓" to denote convergence from the right.

is agreement that NE refinements can play an important role even in two-player zero-sum games, prior algorithms do not scale in practice.[4]

## 1.2 Our contributions

We design a practical algorithm that works in an iterative fashion. At each iteration, it halves the value of $\epsilon$ used in the previous iteration, calls an LP oracle, and checks whether the basis of the solution obtained is also a basis of an optimal solution when $\epsilon \downarrow 0$. The crucial technical contribution is the design of an efficient numerical algorithm for the basis check step. We prove that our practical algorithm requires only polynomial time even in the worst case (we prove that the maximum number of iterations is polynomial and that each iteration runs in polynomial time).

Unlike in prior papers that propose algorithms for QPEs [Miltersen and Sørensen, 2010] and EFPEs [Farina and Gatti, 2017], which were purely theoretical, we show that our algorithm works well in practice also. We apply it to finding a QPE and an EFPE in many different card games: Kuhn poker, Leduc poker with various numbers of ranks, and two versions of Goofspiel [Ross, 1971] with various numbers of ranks. Our algorithm dramatically outperforms the prior algorithms in the literature. It is able to solve games up to four orders of magnitude larger than those previously solvable. It solves games with tens of thousands of nodes in a few minutes, thus showing, for the first time, that trembling-hand refinements can be effectively used in practice—while having a theoretical guarantee of correctness and polynomial run time.

## 2 Extensive-form games and Nash equilibria

Extensive-form games are a general standard representation of games, which can capture sequential and simultaneous moves as well as private information. It is a tree form game supplemented with *information sets*. Each information set contains a set of tree nodes, which are the set of nodes that the player whose turn it is to move cannot distinguish among. We will focus on extensive-form games with perfect recall, that is, where no player forgets what the player knew earlier. Additional background material can be found in the textbook by Shoham and Leyton-Brown [2008].

Extensive-form games with perfect recall are often studied in a tabular representation called the *sequence form* [Romanovskii, 1962; Koller *et al.*, 1996; von Stengel, 1996]. It provides a concise representation that uses space linear in the size of the game tree. A *sequence* $q$ for player $i$ is a subset of $A$ that specifies player $i$'s actions on the path from the root to a decision node $v$. We denote with $Q_i$ the set of sequences of player $i$. A strategy defined on the sequence form is called a *realization plan*. It is a non-negative vector $\mathbf{r}_i$ that maps each sequence $q \in Q_i$ to its probability of being played. A realization plan $\mathbf{r}_i$ for player $i$ is well-defined when it satisfies the linear constraint $\mathbf{F}_i \mathbf{r}_i = \mathbf{f}_i$, where $\mathbf{F}_i$ is a $(|H_i| + 1) \times |Q_i|$ matrix that contains $\{0, \pm 1\}$ entries only, $\mathbf{f}_i$ is a $(|H_i| + 1)$-dimensional vector, and $\mathbf{r}_i$ is the vector that contains the realization plan of player $i$. Finally, the utility function of player $i$ is represented as a sparse matrix $\mathbf{U}_i$ defined only for the profiles of terminal sequences leading to a leaf. The expected payoff for player $i \in \{1, 2\}$ when the two players play according to the realization plan $(\mathbf{r}_1, \mathbf{r}_2)$ is $\mathbf{r}_i^\top \mathbf{U}_i \mathbf{r}_{-i}$ (as customary, we let $-i$ denote the opponent of player $i$).[5]

A NE is a strategy profile in which the strategy of each player is a best response to the strategies of the opponents. In a two-player game, an NE can be defined in the sequence form as a strategy profile $(\mathbf{r}_1^*, \mathbf{r}_2^*)$ where $\mathbf{r}_i^* \in \arg\max \mathbf{r}_i^\top \mathbf{U}_i \mathbf{r}_{-i}$ for all $i \in \{1, 2\}$. Even in zero-sum games, that is games where the sum of the players' payoffs in every leaf is zero, NEs can be unsatisfactory as they do not preclude suboptimal play in branches of the game tree that are not reached in equilibrium, but that might still be reached if players (e.g., humans) can make mistakes. See, for instance, the work by Miltersen and Sørensen [2006] for a discussion of this issue in the context of computer poker.

# 3 Trembling-hand refinements

Nash equilibrium refinement solution concepts curtail the set of Nash equilibria by imposing additional desiderata on the solution. A given equilibrium refinement concept selects some subset of a game's Nash equilibria, thus potentially filtering out some or all of the equilibrium points that exhibit undesirable behaviors. As discussed in the introduction, trembling-hand refinements are a key form of equilibrium refinement. In the next subsections we review key properties of the two main trembling-hand refinements, QPEs and EFPEs. A detailed derivation of the QPE and EFPE formulations is provided in Appendix A.1 and Appendix A.2, respectively.

## 3.1 Quasi-perfect equilibria (QPEs)

In a QPE [van Damme, 1984], a player plays optimally at every information set taking into account possible future mistakes of her opponent but assuming that she will not make future mistakes. Miltersen and Sørensen [2010] show that at least one QPE can be found by forcing the realization $r_i(q)$ of every sequence $q$ to be at least $\epsilon^d$, where $d$ is the depth of the sequence itself. This corresponds to a constraint of the form $\mathbf{r}_i \geq \mathbf{l}_i(\epsilon)$, where $\mathbf{l}_i(\epsilon) > 0$ collects all the lower bounds on the sequence realizations. So, the following result by Miltersen and Sørensen [2010, Equation (25)] holds.

**Proposition 1.** *In a zero-sum extensive-form game with perfect recall, a limit point as $\epsilon \downarrow 0$ of solutions of the linear problem in Figure 1(a) is the strategy of player $i \in \{1, 2\}$ in a QPE.*

$$
\left\{
\begin{aligned}
\max_{\tilde{\mathbf{r}}_i, \mathbf{v}_{-i}, \mathbf{z}_{-i}} \quad & \mathbf{f}_{-i}^\top \mathbf{v}_{-i} + \mathbf{l}_{-i}^\top(\epsilon)\,\mathbf{z}_{-i} \\
\text{s.t.} \quad & \text{①} \quad \mathbf{F}_{-i}^\top \mathbf{v}_{-i} - \mathbf{U}_i^\top \tilde{\mathbf{r}}_i \leq \\
& \qquad\qquad \mathbf{U}_i^\top \mathbf{l}_i(\epsilon) - \mathbf{z}_{-i} \\
& \text{②} \quad \mathbf{F}_i \tilde{\mathbf{r}}_i = \mathbf{f}_i - \mathbf{F}_i\,\mathbf{l}_i(\epsilon) \\
& \text{③} \quad \tilde{\mathbf{r}}_i, \mathbf{z}_{-i} \geq \mathbf{0}
\end{aligned}
\right.
\qquad
\left\{
\begin{aligned}
\max_{\tilde{\mathbf{r}}_i, \mathbf{v}_{-i}} \quad & \mathbf{f}_{-i}^\top \mathbf{v}_{-i} \\
\text{s.t.} \quad & \text{①} \quad \mathbf{R}_{-i}^{-\top}(\epsilon)\mathbf{F}_{-i}^\top \mathbf{v}_{-i} \leq \\
& \qquad\qquad \mathbf{R}_{-i}^{-\top}(\epsilon)\mathbf{U}_i^\top \mathbf{R}_i^{-1}(\epsilon)\tilde{\mathbf{r}}_i \\
& \text{②} \quad \mathbf{F}_i \mathbf{R}_i^{-1}(\epsilon)\tilde{\mathbf{r}}_i = \mathbf{f}_i \\
& \text{③} \quad \tilde{\mathbf{r}}_i \geq \mathbf{0}.
\end{aligned}
\right.
$$

(a) QPE formulation  (b) EFPE formulation

Figure 1: Linear programming formulations of trembling-hand refinements.

## 3.2 Extensive-form perfect equilibria (EFPEs)

In an EFPE [Selten, 1975], a player takes not only the opponent's but also her own possible future mistakes into account. Like QPEs, EFPEs impose a lower bound on the realization of every sequence. Specifically, given a sequence $q$ and an extension with action $a$ (i.e., the sequence $qa$), the realization $r_i(qa)$ has to satisfy the lower bound $r_i(qa) \geq \epsilon_q\, r_i(q)$, where $\epsilon_q > 0$ is a real constant (different sequences can have a different value for $\epsilon_q$). In this paper, we will use a uniform perturbation over the actions of the agent form: $\epsilon_q = \epsilon$ for all sequences $q$. An EFPE with such a uniform perturbation always exists. This allows to express all constraints of the form $r_i(qa) \geq \epsilon\, r_i(q)$ more concisely as $\mathbf{R}_i(\epsilon)\mathbf{r}_i \geq \mathbf{0}$, where $\mathbf{R}_i(\epsilon)$ is the *behavioral* perturbation matrix. With that, the following result was proven by Farina and Gatti [2017].

**Proposition 2.** *In a zero-sum extensive-form game with perfect recall, a limit point as $\epsilon \downarrow 0$ of solutions of the linear problem in Figure 1(b) is the strategy of player $i \in \{1, 2\}$ in an EFPE.*

# 4 Trembling linear programs and their limit solutions

The previous section shows that the problems of finding a QPE and that of finding an EFPE are similar: in both cases an LP with a parameter $\epsilon > 0$ is given, and a limit point of a sequence of optimal solutions to the LP as $\epsilon \downarrow 0$ is sought. We formalize this observation in the concept of a *trembling linear program* (TLP), that is a function

$$
\epsilon \mapsto P(\epsilon) : \left\{
\begin{aligned}
\max \quad & \mathbf{c}(\epsilon)^\top \mathbf{x} \\
\text{s.t.} \quad & \mathbf{A}(\epsilon)\,\mathbf{x} = \mathbf{b}(\epsilon) \\
& \mathbf{x} \geq \mathbf{0}
\end{aligned}
\right.
$$

where $P(\epsilon)$ is an LP, and $\mathbf{A}(\epsilon), \mathbf{b}(\epsilon), \mathbf{c}(\epsilon)$ are polynomials in $\epsilon$, with rational coefficients. Furthermore, we require that the set of all feasible solutions for $P(\epsilon)$ be non-empty for all positive reals

$\epsilon \leq \tilde{\epsilon}$, and that the set of all feasible solutions for $P(0)$ be non-empty and bounded. These additional assumptions are satisfied by the formulations in Figure 1.

**Definition 3** (Limit solution to a TLP). *A* limit solution *to a TLP* $\epsilon \mapsto P(\epsilon)$ *is a limit point, as* $\epsilon \downarrow 0$, *of optimal solutions for* $P(\epsilon)$.

In the case of the QPE TLP (Figure 1(a)), the perturbation variable $\epsilon$ only appears in $\mathbf{b}$; limit solutions to the QPE TLP are QPEs. In the case of the EFPE formulation (Figure 1(b)), the perturbation $\epsilon$ only appears in $\mathbf{A}$; limit solutions to the EFPE TLP are EFPEs.

We now introduce the concept of *basis stability* for TLPs. First, recall that a *basis* of an LP is a subset of the program's variables such that when only those columns of matrix $\mathbf{A}$ that correspond to those variables are included in a new matrix $\mathbf{A}'$, the new matrix $\mathbf{A}'$ is invertible [Bertsimas and Tsitsiklis, 1997, page 55].

**Definition 4** (Stable basis). *Let* $\epsilon \mapsto P(\epsilon)$ *be a TLP. The LP basis* $\mathcal{B}$ *is called* stable *if there exists* $\bar{\epsilon} > 0$ *such that* $\mathcal{B}$ *is optimal for* $P(\epsilon)$ *for all* $\epsilon : 0 < \epsilon \leq \bar{\epsilon}$.

We prove that there is a tight connection between a stable basis and a TLP limit solution. In particular, given a stable basis, one can find a TLP limit solution in polynomial time (all proofs are presented in Appendix A):

**Theorem 5.** *Let* $\mathcal{P} : \epsilon \mapsto P(\epsilon)$ *be a TLP, and let* $\mathcal{B}$ *be a stable basis for* $\mathcal{P}$, *optimal for all* $\epsilon : 0 < \epsilon \leq \bar{\epsilon}$. *Furthermore, let* $\mathbf{x}(\epsilon)$ *be the optimal basic solution of* $P(\epsilon)$ *corresponding to* $\mathcal{B}$. *Then,* $\tilde{\mathbf{x}} = \lim_{\epsilon \downarrow 0} \mathbf{x}(\epsilon)$ *exists, and* $\tilde{\mathbf{x}}$ *is a limit solution to the TLP* $\mathcal{P}$.

Given a TLP and a stable basis $\mathcal{B}$ for it, let $\mathbf{B}(\epsilon)$ denote the basis matrix corresponding to $\mathcal{B}$ in the underlying perturbed LP $P(\epsilon)$, and let $\mathbf{c}_{\mathcal{B}}$ denote the portion of the objective vector $\mathbf{c}$ corresponding to the basic variables. Similarly, let $\bar{\mathbf{B}}(\epsilon)$ and $\mathbf{c}_{\bar{\mathcal{B}}}$ denote the matrix formed by all nonbasic columns and the vector formed by the objective coefficients of all nonbasic variables, respectively.

**Theorem 6.** *Given a TLP* $\epsilon \mapsto P(\epsilon)$, *a basis* $\mathcal{B}$ *is stable if and only if there exists* $\bar{\epsilon} > 0$ *such that*

$$\mathbf{t}_{\mathcal{B}}(\epsilon) := \begin{pmatrix} \mathbf{B}^{-1}(\epsilon)\,\mathbf{b}(\epsilon) \\ \bar{\mathbf{B}}^{\top}(\epsilon)\mathbf{B}^{-\top}(\epsilon)\mathbf{c}_{\mathcal{B}} - \mathbf{c}_{\bar{\mathcal{B}}} \end{pmatrix} \geq \mathbf{0} \qquad \forall \epsilon : 0 \leq \epsilon \leq \bar{\epsilon}.$$

*The vector* $\mathbf{t}_{\mathcal{B}}(\epsilon)$ *is called the* optimality certificate *for* $\mathcal{B}$.

Theorem 6 is a key step in proving the following.

**Theorem 7.** *Given as input a TLP* $\epsilon \mapsto P(\epsilon)$, *there exists* $\epsilon^* > 0$ *such that for all* $0 < \bar{\epsilon} \leq \epsilon^*$, *any optimal basis for the numerical LP* $P(\bar{\epsilon})$ *is stable. Furthermore, such a value* $\epsilon^*$ *can be computed in polynomial time in the input size, assuming that a polynomial of degree* $d$ *requires* $\Omega(d)$ *space in the input.*

## 5 A practical algorithm for finding a TLP limit solution

We now develop a practical algorithm for finding a limit solution in a TLP $\epsilon \mapsto P(\epsilon)$. It avoids the pessimistically small numerical perturbation $\epsilon^*$ of Theorem 7 by using an efficient stability-checking oracle for checking if a basis is stable or not. It enables an iterative algorithm that repeatedly picks a numerical perturbation $\bar{\epsilon}$, computes an optimal basis for the perturbed LP $P(\bar{\epsilon})$, and queries the basis-stability oracle. If the basis is not stable, the algorithm concludes that the perturbation value $\bar{\epsilon}$ was too optimistic, and a new iteration is performed with a smaller perturbation. On the other hand, if the basis is stable, the algorithm takes the limit of the LP solution and returns it as the limit solution of the TLP (by Theorem 5, this is guaranteed to provide a limit solution). Termination of the algorithm is guaranteed by the following observation.

**Observation 8.** *Any value of* $\bar{\epsilon}$ *in the range* $(0, \epsilon^*]$ *guarantees termination of the algorithm. Indeed, by Theorem 7, any optimal basis for* $P(\bar{\epsilon})$ *is stable and makes our iterative algorithm terminate. Furthermore, if after every negative stability test the value of* $\bar{\epsilon}$ *is reduced by a constant multiplicative factor (e.g., halved), then since* $\epsilon^*$ *only has a polynomial number of bits, the algorithm terminates after trying at most a polynomial number of different values for* $\bar{\epsilon}$.

It is not necessary—and in general not true—that the inverse of $\mathbf{B}(0)$ exist. We start from the simpler case in which $\mathbf{B}^{-1}(0)$ exists (thus ruling out the possibility that the optimality certificate $\mathbf{t}_{\mathcal{B}}$ is not defined in 0) and later move to the general case.

## 5.1 Oracle for non-singular basis matrices

If $\mathbf{B}(0)$ is non-singular, then $\mathbf{B}^{-1}(\epsilon)\,\mathbf{b}(\epsilon)$ and $\bar{\mathbf{B}}^{\top}(\epsilon)\,\mathbf{B}^{-\top}(\epsilon)\,\mathbf{c}_{\mathcal{B}} - \mathbf{c}_{\bar{\mathcal{B}}}$ are analytic functions of $\epsilon$ at $\epsilon = 0$. Thus $\mathbf{t}_{\mathcal{B}}(\epsilon)$ is analytic at $\epsilon = 0$. In other words, each entry $t_i(\epsilon)$ of $\mathbf{t}_{\mathcal{B}}(\epsilon)$ is equal to its Taylor expansion $t_i(\epsilon) = \alpha_{i0} + \frac{\alpha_{i1}}{1!}\epsilon + \frac{\alpha_{i2}}{2!}\epsilon^2 + \frac{\alpha_{i3}}{3!}\epsilon^3 + \cdots$ where $\alpha_{ij} = (d^j t_i(\epsilon)/d\epsilon^j)(0)$ is the $j$-th derivative of $t_i(\epsilon)$ evaluated at $\epsilon = 0$.[6] The sign of $t_i(\epsilon)$ in positive proximity[7] of 0 is the same as the first (i.e., relative to the lowest degree monomial) non-zero coefficient of the expansion of $t_i(\epsilon)$ around 0. In other words, there exists a $\bar{\epsilon} > 0$ such that $t_i(\epsilon)$ has the same sign as the first non-zero derivative of $t_i$ evaluated in 0 for all $0 < \epsilon < \bar{\epsilon}$. If all derivatives are 0, then we conclude that $t_i(\epsilon)$ is identically zero around $\epsilon = 0$.

This suggests a simple algorithm for determining whether $\mathcal{B}$ is stable: we compute its optimality certificate $\mathbf{t}_{\mathcal{B}}(\epsilon)$ and repeatedly differentiate each row until we either determine the sign of that row in positive proximity of 0 or we establish that the row is identically zero. If all the rows happen to be non-negative in positive proximity of 0, then the basis is stable; otherwise, it is not. In order to make the algorithm fast, we need to be able to quickly evaluate $\mathbf{t}_{\mathcal{B}}(\epsilon)$ and its derivatives at 0. This fundamentally reduces to our ability to efficiently compute a Taylor expansion of a function of the form $\mathbf{B}^{-1}(\epsilon)\,\mathbf{H}(\epsilon)$ around $\epsilon = 0$, where $\mathbf{H}$ is a matrix or vector whose entries are polynomial in $\epsilon$. This part of the algorithm assumes that a sparse LU factorization of the numerical basis matrix $\mathbf{B}(0)$ is available; one is easy to compute in polynomial time. Below, we will break the presentation of the algorithm into multiple steps. Since the algorithm described below can be applied to any square matrix $\mathbf{M}(\epsilon)$—not only to a basis matrix $\mathbf{B}(\epsilon)$—with polynomial entries and with nonzero determinant at $\epsilon = 0$, we will use the symbol $\mathbf{M}(\epsilon)$ in place of $\mathbf{B}(\epsilon)$.

**Derivatives of $\mathbf{M}^{-1}(\epsilon)\,\mathbf{H}$.** We start by showing how to efficiently and inductively evaluate derivatives of $\mathbf{M}^{-1}(\epsilon)\,\mathbf{H}$ in 0, where $\mathbf{H}$ is a *constant* matrix or vector. We start with a simple lemma.

**Lemma 9.** *For all $n \geq 1$,*

$$\sum_{i=0}^{n} \binom{n}{i} \frac{d^i \mathbf{M}(\epsilon)}{d\epsilon^i} \frac{d^{n-i} \mathbf{M}^{-1}(\epsilon)}{d\epsilon^{n-i}} = \mathbf{0}.$$

Lemma 9 implies that

$$\mathbf{M}(0) \frac{d^n \mathbf{M}^{-1}}{d\epsilon^n}(0) = -\sum_{i=1}^{n} \binom{n}{i} \frac{d^i \mathbf{M}}{d\epsilon^i}(0) \frac{d^{n-i} \mathbf{M}^{-1}}{d\epsilon^{n-i}}(0).$$

Multiplying by $\mathbf{H}$ and introducing the symbol $\mathbf{D}_n := \frac{d^n \mathbf{M}^{-1}}{d\epsilon^n}(0)\,\mathbf{H}$, we obtain

$$\mathbf{M}(0)\,\mathbf{D}_n = -\sum_{i=1}^{n} \binom{n}{i} \frac{d^i \mathbf{M}}{d\epsilon^i}(0)\,\mathbf{D}_{n-i}.$$

The right hand side is relatively inexpensive to compute, especially when $n$ is small. Indeed, computing $(d^i \mathbf{M}/d\epsilon^i)(0)$ amounts to extracting the coefficients of the monomials of degree $i$ of the polynomial entries in $\mathbf{M}$. This can be done extremely efficiently by reading directly from the perturbed LP constraint matrix $\mathbf{A}$. Therefore, if we inductively assume knowledge of $\mathbf{D}_0, \mathbf{D}_1, \ldots, \mathbf{D}_{n-1}$, we can easily compute $\mathbf{D}_n$ using the precomputed LU factorization of $\mathbf{M}(0)$.

**Derivatives of $\mathbf{M}^{-1}(\epsilon)\,\mathbf{H}(\epsilon)$.** We now turn our attention to the computation of the derivatives of $\mathbf{M}^{-1}(\epsilon)\,\mathbf{H}(\epsilon)$, where $\mathbf{H}(\epsilon)$ can be any matrix or vector with polynomial entries. This case is particularly relevant, as it applies to both the primal-feasibility conditions and the reduced costs.

We introduce the formal symbol $\langle i, j \rangle$ defined over pairs $(i, j) \in \mathbb{N}^2$ as $\langle i, j \rangle := \frac{d^i \mathbf{M}^{-1}}{d\epsilon^i}(0) \frac{d^j \mathbf{H}}{d\epsilon^j}(0)$. By means of the product rule, we have that

$$\frac{d^n (\mathbf{M}^{-1}\,\mathbf{H})}{d\epsilon^n}(0) = \sum_{i=0}^{n} \binom{n}{i} \langle i, n-i \rangle. \tag{1}$$

From the previous section, we know how to compute $\langle i+1, j \rangle$ having access to $\langle 0, j \rangle$, $\langle 1, j \rangle$, ..., $\langle i, j \rangle$. On the other hand, $\langle 0, j \rangle = \mathbf{M}(0)^{-1} \, d^j/d\epsilon^j \, \mathbf{H}(0)$ is easy to compute having access to the LU factorization of $\mathbf{M}(0)$. Therefore, Equation (1) gives an efficient way of expanding $\mathbf{M}^{-1}(\epsilon)\mathbf{H}(\epsilon)$ into its power series. Finally, we address the problem of determining, row by row in the derivative vector in the Taylor series, when it is safe to stop after observing only zero-valued derivatives for some row for a number of iterations (i.e., a number of terms in the Taylor series).

**Lemma 10.** *Consider a TLP $\epsilon \mapsto P(\epsilon)$ where $P(\epsilon)$ has $n$ rows and let $m$ be the maximum degree appearing in the polynomial functions defining $P$. Fixed any basis $\mathcal{B}$, if the first $2nm+1$ derivatives of the $i$-th entry of the optimality certificate $\mathbf{t}_{\mathcal{B}}(\epsilon)$ are all zero, the entry is identically zero.*

Since $2nm + 1$ is a polynomial number in the input size, we conclude that the overall algorithm runs in polynomial time, since it terminates in a polynomial number of steps and each step takes polynomial time. There is a more convenient way of determining whether a given row is 0. It is sufficient to pick a random number $\tilde{\epsilon}$ (for example in $(0, 1)$), and evaluate the rational function $t_i$ at $\tilde{\epsilon}$: if $t_i(\tilde{\epsilon}) = 0$, then $t_i$ is identically zero with probability 1 because of the fundamental theorem of algebra. This is the variant that we use in the experiments later in this paper.

Finally, in some cases we can take theoretically sound shortcuts to further enhance the speed of the algorithm. We provide two examples, and we use the two in our implementation of the algorithm. As the first example, consider a TLP that, for each $\epsilon$, has no objective. In this case, the vector $\mathbf{c}$ is zero. This allows us to avoid considering the reduced-cost conditions of Theorem 6, thereby saving time and space. As the second example, if in the QPE formulation of Proposition 1, the LP constraint matrix $\mathbf{A}$ is constant, meaning that the optimality certificate $\mathbf{b}$ has polynomial entries, it is extremely easy to deal with. Again, we can avoid computing all the derivatives of $\mathbf{B}^{-1}$, with large practical savings of time and space.

## 5.2 Oracle for singular basis matrices

We now show how to deal with a singular $\mathbf{B}(0)$. The core idea of our method is to replace the computation of the Taylor expansion of the optimality certificate around $\epsilon = 0$ with a Laurent expansion, that is a power series at $\epsilon$ where negative exponents are allowed. Lemma 11 provides the key result that enables this process.

**Lemma 11.** *Let $\mathbf{M}(\epsilon)$ be a square matrix with polynomial entries, not all of which are identically zero. Then there exist $k \in \mathbb{N}^+$ and matrices $\tilde{\mathbf{M}}(\epsilon)$ and $\mathbf{T}(\epsilon)$ that have polynomials as entries, with nonsingular $\tilde{\mathbf{M}}(0)$, such that*

$$\mathbf{M}(\epsilon) = \epsilon^k \, \mathbf{T}^{-1}(\epsilon) \, \tilde{\mathbf{M}}(\epsilon), \tag{2}$$

*in proximity of $\epsilon = 0$.*

$\mathbf{B}(\epsilon)$ respects the hypotheses of Lemma 11: its entries are not all identically zero since its determinant is not identically zero. Inverting Equation (2) in proximity of $\epsilon = 0$, we obtain

$$\mathbf{M}^{-1}(\epsilon) = \frac{1}{\epsilon^k} \, \tilde{\mathbf{M}}^{-1}(\epsilon) \, \mathbf{T}(\epsilon).$$

Now, given a matrix or vector with polynomial entries $\mathbf{H}(\epsilon)$, suppose that we seek to expand $\mathbf{M}^{-1}(\epsilon) \cdot \mathbf{H}(\epsilon)$ into its Laurent series. Due to Lemma 11, we can rewrite this product as

$$\mathbf{M}^{-1}(\epsilon) \cdot \mathbf{H}(\epsilon) = \frac{1}{\epsilon^k} \left( \tilde{\mathbf{M}}^{-1}(\epsilon)(\mathbf{T}(\epsilon)\mathbf{H}(\epsilon)) \right),$$

where the equality holds in proximity of $\epsilon = 0$. Since $\tilde{\mathbf{M}}(\epsilon)$ is a square matrix with polynomial entries invertible at $\epsilon = 0$ and $\mathbf{T}(\epsilon)\mathbf{H}(\epsilon)$ is a vector or matrix with polynomial entries, we can leverage the machinery of Section 5.1 to expand $\tilde{\mathbf{M}}^{-1}(\epsilon) \cdot (\mathbf{T}(\epsilon)\mathbf{H}(\epsilon))$ into its Taylor series around $\epsilon = 0$. Multiplying this power series by $\epsilon^{-k}$ gives a Laurent series for $\mathbf{M}^{-1}(\epsilon)\mathbf{H}(\epsilon)$ at $\epsilon = 0$. The above shows how to deal with a singular basis matrix. The rest of the algorithm remains unchanged. Together, Sections 5.1 and 5.2 show that, for every TLP, there exists a polynomial-time basis-stability oracle.

Finally, we deal with the last piece of the algorithm, which is the computation of the limit of optimal solutions $\lim_{\epsilon \downarrow 0} \mathbf{x}(\epsilon) = \lim_{\epsilon \downarrow 0} \mathbf{B}^{-1}(\epsilon)\mathbf{b}(\epsilon)$. This task is easy after having computed the Laurent series expansion of $\mathbf{x}(\epsilon)$ around $\epsilon = 0$ (see Sections 5.1 and 5.2).

# 6 Experiments

The LP oracle we use is GLPK 4.63 [GLPK, 2017]. When $\epsilon \geq 1/500$ we use the finite-precision simplex algorithm provided by GLPK, while for $\epsilon < 1/500$ we use the arbitrary-precision variant, as, from our observations, when $\epsilon < 1/500$ the finite-precision solver is doomed to eventually fail due to numerical instability. We experimentally evaluate the performance of the following four algorithms.

**Exact Nash equilibrium solver**, using an LP oracle with arbitrary-precision arithmetics. We warm start the LP oracle with a NE found by an LP oracle that uses finite-precision arithmetics.

**"NPP solver"** for EFPE [Miltersen and Sørensen, 2010] and for QPE [Farina and Gatti, 2017], using an infinite-precision LP oracle; to improve the efficiency, we warm start the LP oracle with a NE found using an LP oracle with finite-precision arithmetics.

**Symbolic-simplex QPE solver** ("M&S Solver"), proposed by Miltersen and Sørensen [2010] to find a QPE. It is a modified simplex algorithm, where some entries are kept as polynomials. We implemented the algorithm as described in the original paper. However, we modified the pivoting rule from the suggested one (pick any nonbasic variable with positive reduced cost) to the greedy one (pick any nonbasic variable with maximum reduced cost). This greatly improved run time.

**Our proposed practical algorithm** from Section 5 for finding a TLP limit solution. We use an LP oracle with infinite-precision arithmetics; to improve the efficiency, we warm start the first iteration of the algorithm with a NE found by an LP oracle with finite-precision arithmetics, and we warm start each subsequent iteration with the NE returned by the previous iteration. The second iteration is performed with $\epsilon = 1/10$, and $\epsilon$ is halved between subsequent consecutive iteration.

Table 1 lists the games we use to benchmark the algorithms, together with their sizes. All the games are fairly standard in the computational game theory literature. Appendix B includes a detailed description of each game.

| **Game Instance** | | **Nodes** | | | | **Information Sets** | | **Sequences** | |
| --- | --- | --- | --- | --- | --- | --- | --- | --- | --- |
| Description | Acronym | Nature | Leaves | Player 1 | Player 2 | Player 1 | Player 2 | Player 1 | Player 2 |
| Kuhn poker | K | 1 | 30 | 12 | 12 | 6 | 6 | 13 | 13 |
| Simple Leduc poker | SL | 13 | 98 | 44 | 44 | 28 | 28 | 57 | 57 |
| Leduc poker — 3 ranks | L3 | 46 | 1 116 | 387 | 387 | 144 | 144 | 337 | 337 |
| Leduc poker — 5 ranks | L5 | 126 | 5 500 | 1 875 | 1 875 | 390 | 390 | 911 | 911 |
| Leduc poker — 8 ranks | L8 | 321 | 22 936 | 7 752 | 7 752 | 984 | 984 | 2 297 | 2 297 |
| Leduc poker — 9 ranks | L9 | 406 | 32 724 | 11 043 | 11 043 | 1 242 | 1 242 | 2 899 | 2 899 |
| Goofspiel — 3 ranks | G3 | 28 | 216 | 273 | 333 | 273 | 273 | 334 | 334 |
| Goofspiel — 4 ranks | G4 | 1 793 | 13 824 | 17 476 | 21 328 | 17 476 | 17 476 | 21 329 | 21 329 |
| Goofspiel* — 3 ranks | G*3 | 0 | 36 | 46 | 57 | 46 | 46 | 58 | 58 |
| Goofspiel* — 4 ranks | G*4 | 0 | 576 | 737 | 916 | 737 | 737 | 917 | 917 |

Table 1: Tree sizes and acronyms of the test game instances.

Every experiment was repeated 50 times. We summarize the average empirical results in Table 2.

**Exact Nash equilibrium computation**. The impact of the rational simplex iteration is minimal in the case of the (exact) NE solver. This is because the rational simplex is warm started from a finite-precision solution, and in practice this avoids further rational-precision pivoting steps. For example, in L9, the finite-precision LP oracle is responsible for roughly 90% of the total compute time, while in G4, this figure grows to about 98%.

**NPP solvers**. The largest poker games solvable within 6 hours were L5 (QPE case) and SL (EFPE case). The NPP solver is significantly slower than the NE solver. This is because 1) it requires a larger number of pivoting steps, and 2) each pivoting step has a higher cost. Unlike the exact NE computation, additional pivoting steps are needed by the rational simplex to find a QPE or an EFPE, even after warm starting from a Nash equilibrium. These extra pivoting steps need to manipulate extremely small constants due to the values of $\epsilon$, hence introducing a large overhead. For instance, in L5, the order of magnitude of the $\epsilon$ used for QPE is $10^{-5883}$. In the QPE case, these expensive numerical values only appear in the objective function and in the right-hand-side constants. In the

| Game | Nash | QPE | | | | | | EFPE | | | | |
| | Simplex | M&S | NPP | TLP Solver | | | | NPP | TLP Solver | | | |
| | LP Solv. | LP Solv. | LP Solv. | LP Solv. | Oracle | Iters. | Final $\epsilon$ | LP Solv. | LP Solv. | Oracle | Iters. | Final $\epsilon$ |
|---|---|---|---|---|---|---|---|---|---|---|---|---|
| K | 1ms | 78ms | 28ms | 45ms | 35ms | 2 | 1/10 | 14ms | 1ms | 5ms | 2 | 1/10 |
| SL | 4ms | 5.71s | 93ms | 19ms | 82ms | 2 | 1/10 | 3.09s | 5ms | 26ms | 3 | 1/20 |
| L3 | 59ms | 36.35s | 37.21s | 362ms | 1.99s | 11 | 1/5120 | > 6h | 1.56s | 1.15s | 12 | 1/10240 |
| L5 | 372ms | > 6h | 27.81m | 2.61s | 4.43s | 10 | 1/2560 | > 6h | 57.65s | 2.87s | 10 | 1/2560 |
| L8 | 3.35s | > 6h | > 6h | 13.38s | 16.65s | 14 | 1/40960 | > 6h | 3.51m | 1.18m | 17 | 1/327680 |
| L9 | 4.90s | > 6h | > 6h | 30.60s | 21.47s | 15 | 1/81920 | > 6h | 22.66m | 28.43s | 15 | 1/81920 |
| G3 | 33ms | 21.64m | 2.88s | 62ms | 251ms | 2 | 1/10 | 1.93m | 36ms | 117ms | 2 | 1/10 |
| G4 | 1.01m | > 6h | > 6h | 2.93m | 28.02s | 5 | 1/80 | > 6h | 1.46m | 1.01m | 5 | 1/80 |
| G*3 | 5ms | 7.03m | 94ms | 20ms | 78ms | 2 | 1/10 | 266ms | 5ms | 22ms | 2 | 1/10 |
| G*4 | 204ms | > 6h | 4.00m | 588ms | 1.48s | 6 | 1/160 | > 6h | 417ms | 1.83s | 6 | 1/160 |

Table 2: Comparison between different solvers (acronyms are as in Table 1). For **TLP Solver**, 'Iters.': number of different perturbations tried; 'Final $\epsilon$': value of $\epsilon$ in the last iteration of our algorithm; 'LP Solv.' and 'Oracle': total compute time over all the iterations spent by the LP solver and our Basis Stability Oracle, respectively. The total compute time of our algorithm is given by the sum of these two quantities.

EFPE case, they appear in the constraint matrix, and there are qualitatively more of them, making the computation even slower. Accordingly, the EFPE NPP solver scales very poorly.

**M&S solver**. The M&S solver only applies to QPEs. Empirically, it is significantly slower than the NPP solver. The reason is two-fold. First, the method is harder to warm start, as the initial basic solution has to be feasible for all sufficiently small $\epsilon > 0$. We initialize the method according to the suggestion of the authors, but this initial vertex is empirically farther away from the optimal one than a NE, which we use to warm start NPP solvers. Second, the pivoting step is more expensive, as the min-ratio test is substituted with a more sophisticated lexicographic test on polynomial coefficients.

**TLP solver**. Our solver represents a dramatic improvement over the prior state-of-the-art algorithms. It finds a QPE/EFPE in few minutes even on the largest game instances. This is a reduction in runtime by 3–4 orders of magnitude. This breakthrough is mainly due to the fact that, in practice, terminates with an $\epsilon$ that is drastically larger than that required by the NPP algorithms. As shown in Table 2, the final $\epsilon$ used by our solver is never smaller than $10^{-6}$, even for the largest instances. In Kuhn poker, the final $\epsilon$ of our solver is about $10^{-5}$ for QPE, while the $\epsilon$ value used by the prior algorithms is of the order $10^{-42}$. In some cases, our method is able to compute an exact refinement without even using rational arithmetic. Still, our TLP solver is significantly slower than the solver for NE. For instance, on L9, the ratio of the compute times is about 30. Here, the bottleneck is mostly due to LP oracle, while the compute time required by our basis stability oracle is relatively small, and becomes relatively even smaller with increasing problem size.

# 7 Conclusions

We introduced *trembling linear problems (TLPs)*, which are linear programs in which every entry can be subject to an independent perturbation expressed by a polynomial in $\epsilon > 0$. We defined a limit solution to a TLP as any limit point of any sequence of optimal solutions for the perturbed linear program as $\epsilon \downarrow 0$. For game theory, TLPs provide a framework for analyzing and computing important Nash equilibrium refinements based on various forms of trembling-hand perfection in two-player zero-sum games, such as *quasi-perfect equilibria (QPEs)* and *extensive-form perfect equilibria (EFPEs)*. We designed an exact polynomial-time algorithm for finding a limit solution to a TLP that, when applied to finding a QPE or EFPE, outperforms the speed of prior algorithms by several orders of magnitude. Our algorithm quickly solves games with tens of thousands of nodes, thus enabling—for the first time—the use of trembling-hand refinements in practice.

**Acknowledgments.** This material is based on work supported by the National Science Foundation under grants IIS-1718457, IIS-1617590, and CCF-1733556, and the ARO under award W911NF-17-1-0082.

## Footnotes

[1]*Proper equilibria (PEs)*, proposed by Myerson [1978], are a non-empty subset of EFPEs, with the additional requirement that the worse an action is for a player, the lower the agent's tremble probability on that

[4]Some algorithms have been proposed for computing approximate trembling-hand equilibria resorting to regret-minimization techniques [Farina *et al.*, 2017] or to smoothing methods paired with bilinear saddle-point techniques [Kroer *et al.*, 2017]. Those algorithms do not provide any guarantee of finding or approximating actual QPEs or EFPEs. Rather, they provide approximate solutions to approximate solution concepts.

[5]We use the superscript $^\top$ to denote the transpose matrix. Similarly, later in the text we use the superscript $^{-\top}$ to denote the inverse transpose matrix.

[6] Throughout this paper, we define the zeroth derivative $d^0 f/d\epsilon^0$ of $f$ to be $f$ itself.

[7] We say that a property parametrized by $\epsilon$ is true in *positive proximity of* 0 to mean that there exists a $\bar{\epsilon} > 0$ such that the property holds for all $\epsilon : 0 < \epsilon < \bar{\epsilon}$. We say that the property is true in *proximity of* 0 if there exists a $\bar{\epsilon} > 0$ such that the property holds for all $\epsilon : 0 < |\epsilon| < \bar{\epsilon}$.

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
