[Supplementary Material]

# A Proofs

## A.1 QPE computation

**Proposition 1.** *In a zero-sum extensive-form game with perfect recall, a limit point as $\epsilon \downarrow 0$ of solutions of the linear problem in Figure 1(a) is the strategy of player $i \in \{1, 2\}$ in a QPE.*

*Proof.* We start by considering the max-min problem for player $i$,

$$
\begin{cases}
\max_{\mathbf{r}_i} & \begin{cases}
\min_{\mathbf{r}_{-i}} & \mathbf{r}_i^\top \mathbf{U}_i \mathbf{r}_{-i} \\
\text{s.t.} & \text{①} \quad \mathbf{F}_{-i}\mathbf{r}_{-i} = \mathbf{f}_{-i} \\
& \text{②} \quad \mathbf{r}_{-i} \geq \mathbf{l}_{-i}(\epsilon)
\end{cases} \\
\text{s.t.} & \text{③} \quad \mathbf{F}_i\mathbf{r}_i = \mathbf{f}_i \\
& \text{④} \quad \mathbf{r}_i \geq \mathbf{l}_i(\epsilon).
\end{cases}
\tag{3}
$$

Taking the dual of the inner problem, we obtain the following:

$$
\begin{cases}
\max_{\mathbf{r}_i} & \begin{cases}
\max_{\mathbf{v}_{-i}, \mathbf{z}_{-i}} & \mathbf{f}_{-i}^\top \mathbf{v}_{-i} + \mathbf{l}_{-i}(\epsilon)^\top \mathbf{z} \\
\text{s.t.} & \text{①} \quad \mathbf{F}_{-i}^\top \mathbf{v}_{-i} + \mathbf{z} \leq \mathbf{U}_i^\top \mathbf{r}_i \\
& \text{②} \quad \mathbf{z}_{-i} \geq \mathbf{0}
\end{cases} \\
\text{s.t.} & \text{③} \quad \mathbf{F}_i\mathbf{r}_i = \mathbf{f}_i \\
& \text{④} \quad \mathbf{r}_i \geq \mathbf{l}_i(\epsilon).
\end{cases}
$$

By substituting $\tilde{\mathbf{r}}_i = \mathbf{r}_i - \mathbf{l}_i(\epsilon), \tilde{\mathbf{r}}_{-i} = \mathbf{r}_{-i} - \mathbf{l}_{-i}(\epsilon)$ we obtain the formulation in the statement. $\square$

## A.2 EFPE computation

**Lemma 12** (Farina and Gatti [2017]). *Let $R(\epsilon)$ be a behavioral perturbation matrix. Then $R(\epsilon)$ is invertible.*

**Proposition 2.** *In a zero-sum extensive-form game with perfect recall, a limit point as $\epsilon \downarrow 0$ of solutions of the linear problem in Figure 1(b) is the strategy of player $i \in \{1, 2\}$ in an EFPE.*

*Proof.* We start by considering the max-min problem for player $i$,

$$
\begin{cases}
\max_{\mathbf{r}_i} & \begin{cases}
\min_{\mathbf{r}_{-i}} & \mathbf{r}_i^\top \mathbf{U}_i \mathbf{r}_{-i} \\
\text{s.t.} & \text{①} \quad \mathbf{F}_{-i}\mathbf{r}_{-i} = \mathbf{f}_{-i} \\
& \text{②} \quad \mathbf{R}_{-i}(\epsilon)\mathbf{r}_{-i} \geq \mathbf{0}
\end{cases} \\
\text{s.t.} & \text{③} \quad \mathbf{F}_i\mathbf{r}_i = \mathbf{f}_i \\
& \text{④} \quad \mathbf{R}_i(\epsilon)\mathbf{r}_i \geq \mathbf{0}.
\end{cases}
\tag{4}
$$

Since the perturbation matrices $\mathbf{R}_i$ and $\mathbf{R}_{-i}$ are invertible (Lemma 12), we can substitute $\tilde{\mathbf{r}}_i = \mathbf{R}_i(\epsilon)\mathbf{r}_i, \tilde{\mathbf{r}}_{-i} = \mathbf{R}_{-i}(\epsilon)\mathbf{r}_{-i}$, obtaining

$$
\begin{cases}
\max_{\tilde{\mathbf{r}}_i} & \begin{cases}
\min_{\tilde{\mathbf{r}}_{-i}} & \tilde{\mathbf{r}}_i^\top \mathbf{R}_i^{-\top}(\epsilon) \mathbf{U}_i \mathbf{R}_{-i}^{-1}(\epsilon)\tilde{\mathbf{r}}_{-i} \\
\text{s.t.} & \text{①} \quad \mathbf{F}_{-i}\mathbf{R}_{-i}^{-1}(\epsilon)\tilde{\mathbf{r}}_{-i} = \mathbf{f}_{-i} \\
& \text{②} \quad \tilde{\mathbf{r}}_{-i} \geq \mathbf{0}
\end{cases} \\
\text{s.t.} & \text{③} \quad \mathbf{F}_i\mathbf{R}_i^{-1}(\epsilon)\tilde{\mathbf{r}}_i = \mathbf{f}_i \\
& \text{④} \quad \tilde{\mathbf{r}}_i \geq \mathbf{0}.
\end{cases}
$$

Taking the dual of the inner problem introducing the vector of dual variables $\mathbf{v}_{-i}$ for constraint ①, we obtain the LP in the statement. $\square$

### A.3 Trembling linear program

**Theorem 5.** *Let $\mathcal{P} : \epsilon \mapsto P(\epsilon)$ be a TLP, and let $\mathcal{B}$ be a stable basis for $\mathcal{P}$, optimal for all $\epsilon : 0 < \epsilon \leq \bar{\epsilon}$. Furthermore, let $\mathbf{x}(\epsilon)$ be the optimal basic solution of $P(\epsilon)$ corresponding to $\mathcal{B}$. Then, $\tilde{\mathbf{x}} = \lim_{\epsilon \downarrow 0} \mathbf{x}(\epsilon)$ exists, and $\tilde{\mathbf{x}}$ is a limit solution to the TLP $\mathcal{P}$.*

*Proof.* The fact that $\tilde{\mathbf{x}}$ is a solution to $\mathcal{P}$ follows directly from Definition 3. Therefore, it is enough to show the existence of $\tilde{\mathbf{x}}$.

For all $\epsilon$, let $\mathbf{B}(\epsilon)$ be the basis matrix in $P(\epsilon)$, corresponding to the given basis $\mathcal{B}$. By definition of stable basis, $\mathcal{B}$ is optimal for $P(\epsilon)$ for all $\epsilon : 0 < \epsilon \leq \bar{\epsilon}$. Therefore, $\mathbf{B}(\epsilon)$ is invertible for all $\epsilon : 0 < \epsilon \leq \bar{\epsilon}$ and we conclude that $\det \mathbf{B}(\epsilon)$ is not identically zero over that range. This implies that $\mathbf{x}(\epsilon) = \mathbf{B}^{-1}(\epsilon)\,\mathbf{b}(\epsilon)$ is well defined and is a vector of rational functions. This, together with the boundedness assumption in Section 4, is enough to conclude that $\lim_{\epsilon \downarrow 0} \mathbf{x}(\epsilon)$ exists. $\qquad\square$

**Theorem 6.** *Given a TLP $\epsilon \mapsto P(\epsilon)$, a basis $\mathcal{B}$ is stable if and only if there exists $\bar{\epsilon} > 0$ such that*

$$\mathbf{t}_{\mathcal{B}}(\epsilon) := \begin{pmatrix} \mathbf{B}^{-1}(\epsilon)\,\mathbf{b}(\epsilon) \\ \bar{\mathbf{B}}^{\top}(\epsilon)\mathbf{B}^{-\top}(\epsilon)\mathbf{c}_{\mathcal{B}} - \mathbf{c}_{\bar{\mathcal{B}}} \end{pmatrix} \geq \mathbf{0} \qquad \forall\,\epsilon : 0 \leq \epsilon \leq \bar{\epsilon}.$$

*The vector $\mathbf{t}_{\mathcal{B}}(\epsilon)$ is called the* optimality certificate *for $\mathcal{B}$.*

*Proof.* Remember that $\mathbf{B}(\epsilon)$ be the basis matrix corresponding to $\mathcal{B}$ in the underlying perturbed LP $P(\epsilon)$. From the theory of LPs, we know that $\mathcal{B}$ is optimal for $P(\epsilon)$ if and only if (see, for instance, the book by Dantzig and Thapa [2006]):

- it is primal-feasible, that is, $\mathbf{B}^{-1}(\epsilon)\,\mathbf{b}(\epsilon) \geq \mathbf{0}$, and

- the reduced costs of all nonbasic columns are nonpositive, that is, $\mathbf{c}_{\bar{\mathcal{B}}}^{\top} - \mathbf{c}_{\mathcal{B}}^{\top}\mathbf{B}^{-1}(\epsilon)\bar{\mathbf{B}}(\epsilon) \leq \mathbf{0}$, where $\mathbf{c}_{\mathcal{B}}$ is the part of $\mathbf{c}$ corresponding to the basic variables, $\mathbf{c}_{\bar{\mathcal{B}}}$ is the part of $\mathbf{c}$ corresponding to the nonbasic variables, $\bar{\mathbf{B}}(\epsilon)$ is the matrix formed by all nonbasic columns.

The *optimality certificate* collects the conditions above into a vector $\mathbf{t}_{\mathcal{B}}(\epsilon)$, which is nonnegative if and only if $\mathcal{B}$ is an optimal basis for the LP $P(\epsilon)$. Therefore, by the definition of basis stability, a basis is stable if $\mathbf{t}_{\mathcal{B}}(\epsilon)$ is nonnegative for all sufficiently small values of $\epsilon$. $\qquad\square$

**Theorem 7.** *Given as input a TLP $\epsilon \mapsto P(\epsilon)$, there exists $\epsilon^* > 0$ such that for all $0 < \bar{\epsilon} \leq \epsilon^*$, any optimal basis for the numerical LP $P(\bar{\epsilon})$ is stable. Furthermore, such a value $\epsilon^*$ can be computed in polynomial time in the input size, assuming that a polynomial of degree $d$ requires $\Omega(d)$ space in the input.*

Before showing the proof of Theorem 7 we introduce a couple of simple facts.

**Lemma 13** (Farina and Gatti [2017])**.** *Let $p(\epsilon) = a_0 + a_1\epsilon + \cdots + a_n\epsilon^n$ be a polynomial over $\mathbb{R}$ such that $a_0 \neq 0$, and let $M = \max_i |a_i|$. Then $p(\epsilon)$ has the same sign of $a_0$ for all $0 \leq \epsilon \leq \epsilon^*$ where $\epsilon^* = |a_0|/(M + |a_0|)$.*

**Lemma 14** (Farina and Gatti [2017])**.** *Let*

$$p(\epsilon) = \frac{a_0 + a_1\epsilon + \cdots + a_n\epsilon^n}{b_0 + b_1\epsilon + \cdots + b_m\epsilon^m}$$

*be a rational function with integer coefficients, where the denominator is not identically zero; let $\mu_a = \max_i |a_i|$, $\mu_b = \max_i |b_i|$, $\mu = \max\{\mu_a, \mu_b\}$ and $\epsilon^* = 1/(2\mu)$. Exactly one of the following holds:*

- $p(\epsilon^*) = 0$ *for all* $\epsilon : 0 < \epsilon \leq \epsilon^*$;

- $p(\epsilon^*) > 0$ *for all* $\epsilon : 0 < \epsilon \leq \epsilon^*$;

- $p(\epsilon^*) < 0$ *for all* $\epsilon : 0 < \epsilon \leq \epsilon^*$.

We now proceed with the proof of Theorem 7.

*Proof of Theorem 7.* We extend the arguments made by Miltersen and Sørensen [2010] and Farina and Gatti [2017], and we generalize them to the case of trembling LPs.

Let $\Omega$ be the set of all bases for $P(\epsilon)$ that are optimal for at least one $\epsilon \in \mathbb{R}^{++}$. For any $\mathcal{B} \in \Omega$, we let $\mathbf{t}_{\mathcal{B}}(\delta)$ be the optimality certificate for $\mathcal{B}$ (Theorem 6). All entries of $\mathbf{t}_{\mathcal{B}}(\delta)$ are rational functions in $\delta$; hence, by Lemma 14, we can find a value $\delta_{\mathcal{B}}^* > 0$, such that all entries of $\mathbf{t}_{\mathcal{B}}(\delta)$ keep the same sign on the domain $0 < \delta \leq \delta_{\mathcal{B}}^*$. We now introduce the function $f : \Omega \to \mathbb{R}^{++}$ mapping every $\mathcal{B} \in \Omega$ to the corresponding value of $\delta_{\mathcal{B}}^*$. Since $\Omega$ is finite, $\min f$ exists and is (strictly) positive; this means that any optimal basis for $P(\min f)$ is optimal for all $P(\epsilon)$ where $0 < \epsilon \leq \min f$, and is therefore stable.

In light of the above, we only need to prove that we can compute a lower bound for $\min f$ in polynomial time. We will assume without loss of generality that the objective function is not perturbed and. Furthermore, we will assume without loss of generality that $\mathbf{A}(\epsilon), \mathbf{b}(\epsilon)$ and $\mathbf{c}(\epsilon)$ only contain integer entries (if not, it is enough to multiply all the entries in the LP by the least common multiple of all denominators to satisfy this assumptions). As long as we can prove that the maximum coefficient appearing in $\mathbf{t}_{\mathcal{B}}$ is polynomially large (in the size of the input TLP), the result follows from the bound in Lemma 14.

The entries of the optimality certificate are obtained by composing sums and products of entries from three vectors: the LP matrix $\mathbf{A}(\epsilon)$, the inverse of the basis matrix $\mathbf{B}^{-1}(\epsilon)$, the vector of constants $\mathbf{b}(\epsilon)$ and the objective function coefficients $\mathbf{c}$. Let $M$ be the largest coefficient that appears in $\mathbf{A}(\epsilon), \mathbf{b}(\epsilon)$ and $\mathbf{c}$, and let $m$ be the largest polynomial degree appearing in $\mathbf{A}(\epsilon)$ and $\mathbf{b}(\epsilon)$. We now study the magnitude of the maximum coefficient and the maximum polynomial degree that can appear in $\mathbf{B}(\epsilon)^{-1}$.

Introducing $\mathbf{C}(\epsilon) = \operatorname{cof} \mathbf{B}(\epsilon)$, the cofactor matrix of $\mathbf{B}(\epsilon)$, we can write the well-known identity

$$\mathbf{B}^{-1}(\epsilon) = \frac{\mathbf{C}^\top(\epsilon)}{\det \mathbf{B}(\epsilon)}$$

*Denominator.* We now give an upper bound on the coefficients of the denominator of the entries in $\mathbf{B}^{-1}(\epsilon)$. By using Hadamard's inequality, we can write

$$\operatorname{coeff}(\det \mathbf{B}(\epsilon)) \leq n^{n/2} M^n \operatorname{coeff}((1 + \epsilon + \cdots + \epsilon^{m_A})^n),$$

where $\operatorname{coeff}(\cdot)$ is the largest coefficient of its polynomial argument. Since $\operatorname{coeff}((1 + \epsilon + \cdots + \epsilon^{m_A})^n) \leq (m_A + 1)^n$, we have

$$\operatorname{coeff}(\det \mathbf{B}(\epsilon)) \leq n^{n/2}((m_A + 1)M)^n,$$
$$\deg(\det \mathbf{B}(\epsilon)) \leq n \cdot m_A.$$

Notice that this bound is valid for all possible basis matrices $\mathbf{B}(\epsilon)$.

*Numerator.* It is easy to see that the bounds on $\operatorname{coeff}(\det \mathbf{B}(\epsilon))$ hold for the cofactor matrix as well:

$$\operatorname{coeff}(\det \mathbf{C}^\top(\epsilon)) \leq n^{n/2}((m_A + 1)M)^n,$$
$$\deg(\det \mathbf{C}^\top(\epsilon)) \leq n \cdot m_A.$$

Again, it is worthwhile to notice that this bound is valid for all possible basis matrices $\mathbf{B}(\epsilon)$.

*Optimality certificate.* We have

$$\operatorname{coeff}(\mathbf{C}^\top \mathbf{b}(\epsilon)) \leq \operatorname{coeff}(\bar{\mathbf{B}}^\top(\epsilon) \, \mathbf{C}^\top(\epsilon) \, \mathbf{c}_{\mathcal{B}}) \leq n^{n/2}((m_A + 1)M)^n \cdot m_A \cdot M_c M,$$
$$\operatorname{coeff}(\det \mathbf{B}(\epsilon) \, \mathbf{c}_{\bar{\mathcal{B}}}) \leq n^{n/2}((m_A + 1)M)^n \cdot M_c.$$

Hence,

$$\operatorname{coeff}(\mathbf{t}_{\mathcal{B}}(\epsilon)) \leq n^{n/2}((m_A + 1)M)^n \cdot M_c(m_A M + 1)$$
$$\leq n^{n/2}((m_A + 1)M)^{n+1} \cdot M_c.$$

Therefore, all coefficients involved require a polynomial number of bits to be represented, concluding the proof. $\qquad\square$

## A.4 Basis stability oracle

**Lemma 9.** *For all $n \geq 1$,*

$$\sum_{i=0}^{n} \binom{n}{i} \frac{d^i \mathbf{M}(\epsilon)}{d\epsilon^i} \frac{d^{n-i} \mathbf{M}^{-1}(\epsilon)}{d\epsilon^{n-i}} = \mathbf{0}.$$

*Proof.* The statement is equivalent to the expansion of the identity

$$d^n/d\epsilon^n (\mathbf{M}(\epsilon) \, \mathbf{M}^{-1}(\epsilon)) = \mathbf{0},$$

true for all $n \geq 1$, by means of the product rule of derivatives. $\square$

**Lemma 15.** *Let $f(\epsilon)/g(\epsilon)$ be a rational function with $g(0) \neq 0$ and with $\deg f = d$. If the first $d+1$ derivatives (starting from the zeroth derivative) of $f(\epsilon)/g(\epsilon)$ evaluated at $\epsilon = 0$ are zero, then $f(\epsilon)$ (and therefore also $f(\epsilon)/g(\epsilon)$) is identically zero.*

*Proof.* We prove the results by induction. The base case ($d = 0$) is clear. Otherwise, let $\deg f = d + 1$, $d \geq 0$. We can write

$$\frac{f(\epsilon)}{g(\epsilon)} = \frac{f_d(\epsilon)}{g(\epsilon)} + a_{d+1} \frac{\epsilon^{d+1}}{g(\epsilon)},$$

where $f_d(\epsilon)$ is a polynomial of degree $d$. All derivatives of order $\neq d+1$ of $\epsilon^{d+1}/g(\epsilon)$ are 0. Hence, since the first $d+2$ derivatives (and thus, in particular, the first $d+1 = 1 + \deg f_d$) of $f/g$ are 0, by induction we deduce that $f_d$ is identically 0.

Therefore, in order to conclude, it suffices to show that the derivative of order $d+1$ of $\epsilon^{d+1}/g(\epsilon)$ is nonzero. To this end, notice that

$$\left( \frac{d^{n+1}}{d\epsilon^{n+1}} \frac{\epsilon^{n+1}}{g(\epsilon)} \right)(0) = \sum_{i=0}^{n+1} \left( \frac{d^i (x^{n+1})}{d\epsilon^i}(0) \cdot \frac{d^{n+1-i}(1/g)}{d\epsilon^{n+1-i}}(0) \right) = (n+1)! \frac{1}{g(0)} \neq 0.$$

$\square$

**Lemma 10.** *Consider a TLP $\epsilon \mapsto P(\epsilon)$ where $P(\epsilon)$ has $n$ rows and let $m$ be the maximum degree appearing in the polynomial functions defining $P$. Fixed any basis $\mathcal{B}$, if the first $2nm + 1$ derivatives of the $i$-th entry of the optimality certificate $\mathbf{t}_{\mathcal{B}}(\epsilon)$ are all zero, the entry is identically zero.*

*Proof.* First of all, notice that the maximum degree that can appear in the denominator of any entry in the optimality certificate $\mathbf{t}_{\mathcal{B}}(\epsilon)$ is $2mn$. We use Lemma 15 to conclude. $\square$

**Lemma 11.** *Let $\mathbf{M}(\epsilon)$ be a square matrix with polynomial entries, not all of which are identically zero. Then there exist $k \in \mathbb{N}^+$ and matrices $\tilde{\mathbf{M}}(\epsilon)$ and $\mathbf{T}(\epsilon)$ that have polynomials as entries, with nonsingular $\tilde{\mathbf{M}}(0)$, such that*

$$\mathbf{M}(\epsilon) = \epsilon^k \, \mathbf{T}^{-1}(\epsilon) \, \tilde{\mathbf{M}}(\epsilon), \tag{2}$$

*in proximity of $\epsilon = 0$.*

*Proof.* We prove the lemma by induction on the number of roots in 0 of $\det \mathbf{M}(\epsilon)$. This number corresponds to the maximum integer $d \geq 0$ such that $\epsilon^d$ is a divisor of $\det \mathbf{M}(\epsilon)$.

*Base case.* When $d = 0$, $\det \mathbf{M}(0) \neq 0$, and therefore $\mathbf{M}(0)$ is nonsingular. The result holds trivially by letting $\tilde{\mathbf{M}}(\epsilon) = \mathbf{M}(\epsilon)$ and $\mathbf{T}(\epsilon) = \mathbf{I}$ be the identity function for all $\epsilon$.

*Inductive step.* Suppose the results holds for all matrices $\mathbf{M}(\epsilon)$ whose determinants have $d \leq \bar{d} - 1$ roots in 0, with $\bar{d} \geq 1$. We will now show that the results holds when $d = \bar{d}$ as well. Since $\bar{d} \geq 1$, $\det \mathbf{M}(0) = 0$ and therefore there exists a nonzero vector $\mathbf{v}$ such that $\mathbf{v}^\top \mathbf{M}(0) = \mathbf{0}$. This necessarily means that $\epsilon$ divides all entries of $\mathbf{v}^\top \mathbf{M}(\epsilon)$, and therefore $\epsilon^{-1} \mathbf{v}^\top \mathbf{M}(\epsilon)$ is a vector with polynomial entries. Let $i$ be any index such that $v_i \neq 0$, and consider the new matrix $\mathbf{M}'(\epsilon)$ obtained by substituting the $i$-th row in $\mathbf{M}(\epsilon)$ with $\epsilon^{-1} \mathbf{v}^\top \mathbf{M}(\epsilon)$. It is immediate to verify that we can write this operation compactly as

$$\mathbf{M}'(\epsilon) = \frac{1}{\epsilon} \mathbf{S}(\epsilon) \, \mathbf{M}(\epsilon).$$

| | K | SL | L3 | L4 | L5 | L6 | L7 | L8 | L9 | G3 | G4 | G*3 | G*4 |
|---|---|---|---|---|---|---|---|---|---|---|---|---|---|
| QPE | $10^{-42}$ | $10^{-257}$ | $10^{-1875}$ | $10^{-3520}$ | $10^{-5883}$ | $10^{-9045}$ | $10^{-12966}$ | $10^{-17672}$ | $10^{-23187}$ | $10^{-1799}$ | $10^{-169732}$ | $10^{-276}$ | $10^{-5865}$ |
| EFPE | $10^{-67}$ | $10^{-381}$ | $10^{-2943}$ | $10^{-5611}$ | $10^{-9225}$ | $10^{-13822}$ | $10^{-19434}$ | $10^{-26088}$ | $10^{-33807}$ | $10^{-3215}$ | $10^{-323797}$ | $10^{-441}$ | $10^{-10405}$ |

Table 3: Orders of magnitude of $\epsilon$ used for the NPP solver.

where $\mathbf{S}(\epsilon)$ is a nonzero square matrix with polynomial entries. Hence,

$$\mathbf{M}(\epsilon) = \epsilon\, \mathbf{S}^{-1}(\epsilon)\, \mathbf{M}'(\epsilon). \tag{5}$$

$\mathbf{M}'(\epsilon)$ is a square matrix with polynomial entries not all of which are identically zero; however, the number of roots in 0 of $\det \mathbf{M}'(\epsilon)$ is smaller than $\bar{d}$ since we multiplied one of the rows by $\epsilon^{-1}$. Thus, we can apply our inductive hypothesis to $\mathbf{M}'(\epsilon)$ and write

$$\mathbf{M}'(\epsilon) = \epsilon^k\, \mathbf{T}'^{-1}(\epsilon)\, \tilde{\mathbf{M}}'(\epsilon)$$

for some integer $k \geq 0$. Substituting into Equation (5), we obtain

$$\mathbf{M}(\epsilon) = \epsilon^{k+1}\, \mathbf{S}^{-1}(\epsilon) \mathbf{T}'^{-1}(\epsilon)\, \tilde{\mathbf{M}}'(\epsilon)$$
$$= \epsilon^{k+1}\, (\mathbf{T}'(\epsilon)\, \mathbf{S}(\epsilon))^{-1} \tilde{\mathbf{M}}'(\epsilon).$$

Sine $\mathbf{T}'(\epsilon)\, \mathbf{S}(\epsilon)$ is a square matrix with polynomial entries, the proof is complete. $\qquad\square$

## B   Test game instances

Our testbed includes the following classes of extensive-form games. Table 1 shows the acronyms that we will use, together with tree size results.

**Kuhn poker** [Kuhn, 1950]. This is a simplified poker variant with less than 100 leaves and a three-card deck (King, Queen, and Jack). Each player first puts a payment of 1 into the pot. Each player is then dealt one of the three cards, and the third is put aside unseen. A single round of betting then occurs (with betting parameter $p = 1$, see the description of Leduc poker below). If no player folds, a showdown occurs. The player with the higher card wins the pot; in case of tie, the pot is split evenly.

**Simple Leduc poker**. The deck consists of two kings and two jacks. Each player first puts a payment of 1 into the pot. A private card is dealt to each, followed by a betting round (with betting parameter $p = 2$, see the description of Leduc poker below), then a public card is dealt, followed by another betting round (with $p = 4$, see the description of Leduc poker below). If no player has folded, a showdown occurs. In Simple Leduc poker, a showdown has two possible outcomes: one player has a pair, or both players have the same private card. In the former case, the player with the pair wins the pot. In the latter case, the pot is split evenly among the players.

**Leduc poker** [Southey *et al.*, 2005]. This is a widely-used benchmark in the imperfect-information game-solving community. We test on a larger variant of the game in order to better evaluate scalability. In our enlarged variant, the deck contains $k \geq 3$ card ranks, that is, it consists of pairs of cards $1, \ldots, k$, for a total $2k$ cards. We use $k \in \{2, 3, 4, 5, 6, 7, 8, 9\}$ to generate test games of different sizes. Each player initially pays one chip to the pot, and is dealt a single private card. After a round of betting (with betting parameter $p = 2$, see below), a community card is dealt face up, and a subsequent round of betting (with betting parameter $p = 4$, see below) is played. If no player has folded, a showdown occurs, and both players reveal their private cards. If either player pairs their card with the community card they win the pot. Otherwise, the player with the highest private card wins. In the event that both players have the same private card, they draw and split the pot.

Each round of betting with betting parameter $p$ goes as follows.

1. Player 1 can check or bet $p$. If Player 1 checks, the betting round continues with Step (2); otherwise, the betting round continues with Step (4).

2. Player 2 can check or raise $p$. If Player 2 checks, the betting round ends; otherwise, Player 2 add $p$ to the pot the betting round continues with Step (3).

3. Player 1 can fold or call. If Player 1 folds, Player 2 wins the pot and the game ends; otherwise, Player 1 puts $p$ in the pot the betting round ends.

4. Player 2 can fold or call. If Player 2 folds, Player 1 wins the pot and the game ends; otherwise, Player 2 adds $p$ to the pot and the betting round ends.

**Goofspiel** [Ross, 1971]. We consider a parametric version of the game in order to better test scalability. In our variant with $k$ card ranks, three decks containing cards with values $\{1, 2, \ldots, k\}$ are distributed: one is shuffled and laid face-down on the table, and the two players hold on deck each in their hands. Exactly $k$ turns happen; in each turn, the topmost card in the table deck is revealed for both players to observe. Then, both players simultaneously play a card from their hand. The player with the highest card wins as many points as the value of the revealed table cards; in case of tie, this value is split evenly among the players. The revealed table card and the two played cards are discarded and the a new turn can begin.

**Goofspiel**$^*$. In this variant of Goofspiel, the table deck is not shuffled, but rather is laid face-up on the table with cards $1, 2, \ldots, k$ disposed in this order from top to bottom. The dynamics of the game are left unchanged.