[Reviews · NeurIPS 2018]

Reviewer 1



*** POST-REBUTTAL *** I acknowledge authors response and increase the score as mentioned. *** PRE-REBUTTAL *** This paper presents a new practical algorithm for finding exact solution to trembling-hand equilibrium of zero-sum extensive form games The idea of the paper is nice, while the high-level picture is relatively easy to explain (and authors do a good job in doing so). It boils to having an efficient way of checking whether the basis of LP for a given epsilon is stable - in other words, whether it does not change if we were to decrease the epsilon. Using this oracle, one simply keeps decreasing the epsilon until the epsilon is small enough for the corresponding LP to have a stable basis. Quality + Results and methods well supported by the provided theorems. + Experiments evaluated on nice set of games + Strong empirical results - My main issue is that the authors apparently made an error during the submission process. The supplementary PDF is the same as the submission PDF. This is very unfortunate since the supplementary PDF was supposed to contain proofs to the very key theorems of the paper (e.g. Theorem 5, 6). I am forced to at this point give a reject score for the submission. If authors upload the correct supplementary material with the key proofs, I will happily increase the score to 9 (“excellent submission”) since it is a very nice paper. Clarity + Authors do a good at explaining the idea + Logical flow of the paper is well structured - In the abstract, you should state that the result is about two-player zero-sum games, not just zero-sum games. Originality: + The method is original, and the key idea behind the stable basis for trembling equilibrium seem to be novel + Authors do a good job of separating previous work and how it relates here Significance + Practically increasing the game size for which the computation of the mentioned solution concept is feasible is a significant contribution

Reviewer 2



*** POST-REBUTTAL *** Despite the missing appendices, I had tried to work out the authors' proofs based on the sketches they provided in the main paper: whatever questions I had, they addressed in the rebuttal (I upgraded my assessment from 6 to 7 to indicate just this). As far as I can tell, the authors' analysis is sound and I believe the paper clears the bar for acceptance. *** PRE-REBUTTAL *** In this paper, the authors propose a novel algorithm for calculating trembling-hand equilibria in two-player zero-sum games in extensive form. In a nutshell, the authors reformulate the zero-sum game at hand as a linear program (a standard trick), and they then describe an oracle mechanism for deciding whether a basis of the resulting linear program is stable or not. When such a basis is computed, the algorithm lets the magnitude of the trembles go to zero and ultimately outputs a trembling hand equilibrium. I found the paper to be overall well-written, clearly structured, and the analysis morally sound (at least, to the best of my understanding). At the same time however, there are some important points where I found the authors' presentation unsatisfactory. In particular, I have the following concerns: - A good amount of the authors' analysis relies on the optimality certificate t_B of a given basis being an analytic function of epsilon (the perturbation magnitude). From what I could gather, the polynomial (in epsilon) nature of the LP has to do with the fact that the underlying game is finite (though the specifics of how epsilon enters the various expressions are not particularly well-explained either). However, given that the expression for t_B involves the inverse of B (which is a rational function of its components), it is not clear why the resulting function does not have a pole at 0 (or below the perturbation threshold predicted by Theorem 7 for that matter). This is further compounded by the last sentence of Section 5 which states that "this task is easy after having computed the Laurent [as opposed to Taylor] series of x(epsilon)", which also seems to suggest that t_B is meromorphic - not holomorphic. This is a central point in the authors' reasoning so I would expect a more rigorous and more detailed treatment. - Regarding the algorithm's runtime: the authors claim that it is polynomial, but there's a couple of points that are not clear to me: 1. The lowest non-zero term in the Taylor expansion of t_B might have an exponentially high degree: in that case, the algorithm would need to go through exponentially many derivatives before getting to the order of the expansion. [To be clear: I am not claiming that this is the case, I am just saying that this cannot be a priori excluded.] The authors need to show that this cannot happen, otherwise I fail to see why the algorithm would terminate in polynomial time. 2. The number of bases that need to be checked might also be super-polynomial: why is this not the case? All in all, statements like the one in L244 ("the optimality certificate is extremely easy to deal with") seem too cavalier to me, especially when considering subtle complexity issues as the authors are doing. - The argument in p.2 that "no NE refinement was used [in Libratus] to capitalize on the opponent's mistakes" is vague at best, misleading at worst. Even though I appreciate the algorithmic importance of calculating NE refinements (trembling hand/sub-game perfect, QRE or otherwise), this statement is neither correct nor sufficient to single out the relevance of trembling hand equilibria. The cited book of van Damme (1984) contains many other refinements that coud "capitalize on mistakes" (basically, any Nash equilibrium that is not supported on weakly dominated strategies would do a fine job at exploiting mistakes). All in all, I'm on the fence about this submission: I did like it, but some points could be better explained and I also feel a better venue for it would be a conference like EC, SODA or FOCS.

Reviewer 3



Summary: The paper presents new exact algorithms for computing game theoretic equilibria. These algorithms are based on a new general solution technique for solving perturbed LPs. Empirical evaluation shows that the algorithms can find exact equilibria orders of magnitude faster than the state of the art methods. Quality: It is very hard for me to asses the technical quality of the contribution for two main reasons. 1) The authors unfortunately submitted the very same file as the main submission and the supplementary material. Since the proofs, which are crucial for evaluating the correctness of the algorithm are supposed to be in the supplementary materials, it is impossible to validate them. 2) The paper assumes a detailed knowledge of linear programming, which is, in my opinion, beyond the level that I would expected from an AI researcher. Terms, such as the "optimal basis of an LP" are used without any introduction. It is not specified what is denoted by the bar over a (real) matrix or by the upper index "-T", if it is not a typo. I suggest clearly defining the used symbols, even though they may be standard in a different community, referring to some suitable introductory texts, where the required knowledge can be obtained or ideally including a brief introduction to the necessary concepts in the appendix. The experimental evaluation seems to be sufficient and clearly demonstrate the point of the paper well. Clarity: With the exceptions of sections 4 and 5, the paper is written clearly and it is easy to understand. The key technical sections do not properly introduce the used notation and assumes knowledge of several concepts which cannot be considered common knowledge in AI. Originality: I cannot asses how original is the solution from the linear programming perspective. However, its application for solving Nash equilibrium refinements is certainly novel. Significance: It the algorithm is correct, it allows for a substantial increase in the speed of computation of refinements of Nash equilibria. This is significant mainly from the basic research perspective, as it introduces a new algorithm that can be useful for a wider class of similar problems. On the other hand, I consider the practical significance of the algorithm to be limited. A highly related paper, which is not cited in this work (Čermák et al. 2014), shows that even the undominated equilibrium gets almost the optimal performance against imperfect opponents in practice. This refinement is easy to compute using linear programming. Practical significance of the paper would be greatly improved if it provided some evidence that the proposed refinements can perform substantially better than the undominated equilibrium in practice. Čermák J, Bošanský B, Lisý V. Practical performance of refinements of Nash equilibria in extensive-form zero-sum games. In Proceedings of the Twenty-first European Conference on Artificial Intelligence 2014 Aug 18 (pp. 201-206). After rebuttal: After going through the appendix, I am happy to increase my scores. The algorithm seems to be sound and its speed is impressive. I am still not convinced about practical significance. I am aware of the simple artificial counterexamples showing that the stronger equilibrium refinements make sense. I was questioning whether they have any practical benefit over the easily computable undominated equilibrium in practical problems. There is some evidence that they do not. I believe that showing a substantial improvement on a specific game with a realistic error model for the opponent would make the paper much more significant.